# Alterations in Glucose Metabolism Due to Decreased Expression of Heterogeneous Nuclear Ribonucleoprotein M in Pancreatic Ductal Adenocarcinoma

**DOI:** 10.3390/biology10010057

**Published:** 2021-01-14

**Authors:** Jun-ichi Takino, Takuma Sato, Isamu Hiraishi, Kentaro Nagamine, Takamitsu Hori

**Affiliations:** 1Faculty of Pharmaceutical Sciences, Hiroshima International University, 5-1-1 Hirokoshingai, Kure, Hiroshima 737-0112, Japan; t-sato@hirokoku-u.ac.jp (T.S.); madaozukiorange@gmail.com (I.H.); hori@hirokoku-u.ac.jp (T.H.); 2Faculty of Health Sciences, Hiroshima International University, 5-1-1 Hirokoshingai, Kure, Hiroshima 737-0112, Japan; k-nagami@hirokoku-u.ac.jp

**Keywords:** glucose, metabolism, consumption, heterogeneous nuclear ribonucleoprotein M (HNRNPM), pancreatic ductal adenocarcinoma (PDA)

## Abstract

**Simple Summary:**

Pancreatic cancer has one of the worst prognoses when compared to those of other cancer subtypes. One of the reasons is the resistance of this tumor to the hypovascular environment (an environment with low blood flow and low supply of oxygen and nutrients (especially glucose)). However, the detailed mechanism remains elusive. Recently, it has been reported that heterogeneous ribonuclear protein M (HNRNPM) is a splicing factor associated with malignant tumors. Thus, in this study, we investigated the expression and effects of HNRNPM in pancreatic ductal adenocarcinoma (PDA). We revealed that HNRNPM was highly expressed in pancreatic tissues but expression decreased in PDA tissues. Furthermore, we found that knockdown of HNRNPM protein expression under low-glucose conditions altered glucose metabolism and prolonged cell survival by suppressing glucose consumption. These results suggest that reduced expression of HNRNPM in PDAs may be involved in adaptation to a hypovascular environment, and that therapeutic agents for this target may lead to improved prognosis for pancreatic cancer.

**Abstract:**

The prognosis of pancreatic cancer is considerably worse than that of other cancers, as early detection of pancreatic cancer is difficult and due to its hypovascular environment, which involves low blood flow and a low supply of oxygen and nutrients. Moreover, pancreatic cancer demonstrates a mechanism that allows it to survive in a hypovascular environment. However, the detailed mechanism remains elusive. Recently, it has been reported that heterogeneous ribonuclear protein M (HNRNPM) is a splicing factor associated with malignant tumors. Thus, in this study, we investigated the expression and effects of HNRNPM in pancreatic ductal adenocarcinoma (PDA). We observed that HNRNPM expression, which is highly expressed in pancreatic tissues, was reduced in PDA tissues. Additionally, knockdown of HNRNPM under low-glucose conditions that mimic a hypovascular environment was shown to alter glucose metabolism and prolong cell survival by suppressing glucose consumption. These results suggest that the decreased expression of HNRNPM in PDA may be involved in its adaptation to a hypovascular environment.

## 1. Introduction

Pancreatic cancer has one of the worst prognoses when compared to those of other cancer subtypes. In the USA, the 5-year relative survival rate for pancreatic cancer patients of all stages combined is 9%. Moreover, the majority of patients are diagnosed at a late stage (progression to metastasis), for which the 5-year survival rate is 3%. Furthermore, it was estimated that approximately 58,000 people would be diagnosed in 2020 with pancreatic cancer and 47,000 people would succumb to the disease, with pancreatic cancer projected as the third leading cause of cancer-related death [1].

One of the reasons for the low survival rate of pancreatic cancer is that the symptoms peculiar to pancreatic cancer rarely appear and the pancreas is surrounded by various organs, making early detection and take cancer cells difficult [2,3]. Therefore, pancreatic cancer is usually diagnosed at an advanced stage and systemic therapy is central to treatment. So far, targeted therapy alone (except for some gene therapy) or in combination with standard cytotoxic therapy has proven to be disappointing [4]. In addition to these challenges, this tumor is generally found in a hypovascular environment (an environment with low blood flow and low supply of oxygen and nutrients (especially glucose)), unlike tumors in other types of organs [3,4]. This renders anti-cancer drugs less effective, as the tumor cannot be easily accessed by the drugs. Moreover, this also indicates that the tumor has developed mechanisms which increases its survival in this hypovascular environment [3,5,6]. Therefore, to improve the survival rate of pancreatic cancer patients, the development of biomarkers that can help with early detection, as well as therapeutic drugs that can counter this survival mechanism in a hypovascular environment is necessary. However, the mechanism of survival in a hypovascular environment remains to be elucidated.

Heterogeneous nuclear ribonucleoprotein M (HNRNPM) is a splicing factor that plays an important role in the regulation of gene expression by processing heterogeneous nuclear RNAs into mature mRNAs [7]. It has been reported that HNRNPM is associated with the development of malignant tumors. An increase in HNRNPM expression was found to be correlated with the pathogenesis of ovarian cancer and poor outcome of colon cancer and Ewing sarcoma [8,9,10,11], as well as invasion and metastasis in breast and colon cancer [12,13,14,15,16]. In contrast, a decrease in HNRNPM expression was shown to be correlated with a more aggressive phenotype in prostate cancer [17]. However, whether HNRNPM plays a role in pancreatic cancer remains unknown.

In this study, we found that HNRNPM is highly expressed in pancreatic tissues and that its expression is reduced in pancreatic ductal adenocarcinoma (PDA) tissues. Furthermore, we demonstrated that a decrease in HNRNPM protein levels prolonged cell survival by suppressing glucose consumption accompanied by altered metabolism in the PDA cell line MIA PaCa-2 under low-glucose conditions.

## 2. Materials and Methods

### 2.1. Chemicals, Human Tissue cDNA, and Cell Line

All chemicals were commercial samples of high purity and were used according to the manufacturer’s instructions. Human multiple tissue cDNA (MTCTM) panel I (K1420-1), containing cDNA from 8 human tissues, was purchased from Clontech Laboratories, Inc. (Palo Alto, CA, USA). Human pancreatic cDNA from patients with diabetes (C1236188Dia) and adenocarcinoma (C1235188-10) was purchased from BioChain Institute, Inc. (Newark, CA, USA). The MIA PaCa-2 cell line (JCRB0070) derived from the PDA of a 65-year-old male was obtained from the JCRB Cell Bank (Osaka, Japan).

### 2.2. Polymerase Chain Reaction (PCR)

The prepared cDNA was amplified using specific primers and the Quick Taq^®^ HS DyeMix (Toyobo Co., Ltd., Osaka, Japan) using a thermal cycler. The primers used were as follows: HNRNPM, 5′-CTG GAT TAT AAA GTT GGC TG-3′, and 5′-TGC CTT CCA TTC CCA TTC CTG-3′; GAPDH, 5′-ATC AAT GAC CCC TTC ATT GAC CTC A-3′, and 5′-TGG TTC ACA CCC ATG ACG AAC ATG G-3′.

### 2.3. Cell Cultures

MIA PaCa-2 cells were grown in high glucose Dulbecco’s modified Eagle medium (DMEM; FUJIFILM Wako Pure Chemical Corporation, Osaka, Japan) supplemented with 10% fetal bovine serum (FBS; Equitech-Bio Inc., Kerrville, TX, USA) under standard cell culture conditions (humidified atmosphere, 5% CO_2_, 37 °C). The cell media was then replaced with low-glucose DMEM supplemented with 10% FBS and cells at a density of 0.8 × 10^5^ cells/mL were seeded in culture dishes and plates (TrueLine, Pittston, PA, USA), and cultured until use in experiments (excluding experiments under high-glucose conditions).

### 2.4. HNRNPM Knockdown

The two Silencer Select siRNAs targeting HNRNPM (si1: SASI_Hs01_00049126; si2: SASI_Hs01_00049128) (Sigma–Aldrich Co. LLC., St. Louis, MO, USA) were used to downregulate the expression of HNRNPM in MIA PaCa-2 cells. A non-targeting siRNA (SIC-001, Sigma-Aldrich Co. LLC.) was used as control. siRNA transfection was performed according to the manufacturer’s instructions. Briefly, the cells were seeded and cultured for 8 h. Thereafter, the cells were transfected with a mixture of the MISSION^®^ siRNA Transfection Reagent (Sigma–Aldrich Co. LLC.) and 8.3 nM siRNA, which was pre-incubated for 10 min.

### 2.5. Preparation of Cell Lysates and Western Blotting Analysis

Cells were lysed using an IP Lysis Buffer containing the Halt Protease and Phosphatase Inhibitor (Thermo Fisher Scientific, Inc., Waltham, MA, USA), centrifuged at 12,000× *g* for 10 min at 4 °C, and then the supernatant was recovered. Protein concentrations were measured using the Bradford assay (Bio-Rad Laboratories, Inc., Hercules, CA, USA). Cell lysates were dissolved in LDS sample buffer containing 10% sample reducing agent (Thermo Fisher Scientific, Inc.), boiled for 10 min at 70 °C, separated by SDS-PAGE, and then electro-transferred onto polyvinylidene difluoride (PVDF) membranes (Merck Millipore, MA, USA). The membranes were then blocked for 1 h at 37 °C using a PVDF blocking reagent for Can Get Signal^®^ (Toyobo Co., Ltd.). After washing with PBS containing 0.05% Tween 20 (PBS-T), the membranes were incubated with a mouse anti-HNRNPM antibody (Merck Millipore, MA, USA, 03-100 at 1:10,000) or mouse anti-β-actin antibody (Santa Cruz Biotechnology, Inc., Dallas, TX, sc-47778 at 1:12,000) diluted in Can Get Signal^®^ Solution 1 (Toyobo Co., Ltd.) for 1 h. Subsequently, the membranes were washed thrice with PBS-T and incubated with anti-mouse IgG antibodies (DakoCytomation A/S, Glostrup, Denmark, P0260 at 1:5000) diluted in Can Get Signal^®^ Solution 2 (Toyobo Co., Ltd.) for 1 h. After five additional washes with PBS-T, immunoreactive proteins were detected using Western BloT Quant HRP or Ultra Sensitive HRP Substrate (Takara Bio, Inc., Shiga, Japan) and Amersham hyperfilm ECL (GE Healthcare Ltd., Buckinghamshire, UK).

### 2.6. Cell Proliferation Assay

After the cells were cultured for 24 to 60 h, a volume of 10 µL/well of the Cell Counting Kit-8 assay solution (Dojindo Laboratories, Kumamoto, Japan) was added and the cells were further incubated for 1 h. Absorbance was then measured at 450 nm and 650 nm using a microplate reader. The net difference in absorbance (A450–A650) was used as a measure of metabolic activity. The metabolic activity of the siRNA control-treated cells was considered to be 100%.

After the cells were cultured for 24 to 60 h, an equal amount of the CellTiter-Glo 2.0 Cell Viability Assay reagent (Promega, Madison, WI, USA) was added to each well and incubated for 10 min. Luminescence was measured using a luminometer to determine the amount of ATP, as well as cell proliferation. The amount of ATP in the siRNA control-treated cells was considered to be 100%.

### 2.7. Glucose and Lactate Concentration in the Medium

After the cells were cultured for 24–48 h (glucose assay) or for 48 h (lactate assay), the medium was collected and the centrifuged supernatant was analyzed using the Glucose kit Glucose C II Test Wako (FUJIFILM Wako Pure Chemical Corporation) for 15 min, or the Glycolysis Cell-Based Assay Kit (Cayman Chemical, Ann Arbor, MI, USA) for 30 min. The absorbance was then measured at 505 nm (glucose assay) or 490 nm (lactate assay) using a microplate reader and the concentration of glucose and lactate in the medium was quantified using a calibration curve. Low glucose DMEM supplemented with 10% FBS has a glucose concentration of approximately 1000 μg/mL and a lactate concentration of 0 mM.

### 2.8. Glucose Uptake

Glucose uptake was measured using the Glucose Uptake-Glo Assay and the CellTiter-Glo 2.0 Cell Viability Assay (Promega). Briefly, the cells were cultured for 48 h and then the medium was removed and washed with PBS (-). Cells were then incubated with 1 mM 2-deoxyglucose (2-DG) in PBS (-) for 10 min, after which the stop and neutralization buffer was added. The accumulated 2-DG-6-phosphate (2DG6P) was converted into a luminescence signal via incubation with a 2DG6P detection reagent for 1 h, and then the luminescence was measured using a luminometer. Similarly, the number of cells was determined using the CellTiter-Glo 2.0 reagent, which quantifies the amount of ATP, and corrected for per-cell glucose uptake. The glucose uptake ratio in the siRNA control-treated cells was considered to be 1.

### 2.9. Statistical Analysis

All experiments were performed in duplicate and repeated at least two or three times; each experiment yielded essentially identical results. Data are expressed as mean ± standard deviation (SD). The statistical significance of the differences between group mean values was determined using a one-way analysis of variance and Student’s *t*-test. A *p*-value of < 0.05 was considered statistically significant.

## 3. Results

### 3.1. HNRNPM Is Highly Expressed in Pancreatic Tissues but Expression Is Reduced in PDA Tissues

The mRNA expression of HNRNPM in both normal and diseased human tissues was analyzed by PCR. We found that HNRNPM was expressed in 6 human tissues, with the exception of the brain and skeletal muscles. Moreover, its expression was found to be the highest in the pancreas when compared with other tissues (Figure 1a). Additionally, HNRNPM mRNA levels, which are high in normal pancreatic tissues, were found to be markedly reduced in cancer patients when compared with healthy subjects and patients with diabetes (Figure 1b).

### 3.2. HNRNPM Knockdown Prolongs Cell Survival of MIA PaCa-2 Cells under Low-Glucose Conditions

We examined the effect of decreased HNRNPM expression on PDA under low-glucose conditions, which mimic a hypovascular environment. HNRNPM protein levels were shown to be significantly reduced in the PDA cell line MIA PaCa-2 cells after treatment with siRNA against HNRNPM for 24 to 60 h (Figure 2a–c).

Next, the proliferation ability and the number of cells were determined by using the Cell Counting Kit-8 assay (metabolic activity), CellTiter-Glo 2.0 Cell Viability Assay (ATP amount), or by cell counting. HNRNPM knockdown cells exhibited significantly increased metabolic activity after 48 and 60 h when compared to control cells (Figure 3a). However, a remarkable decrease in metabolic activity was also observed in control cells after 48 h, as well as in HNRNPM knockdown cells after 60 h. Furthermore, HNRNPM knockdown cells exhibited significantly increased ATP levels under low-glucose conditions after 60 h when compared to control cells (Figure 3b). Additionally, micrographs, which illustrated time-dependent cell proliferation even after 60 h, (Figure 3c) revealed a significantly higher number of HNRNPM knockdown cells under low-glucose conditions after 60 h when compared to control cells (Figure 3d). Therefore, these results indicate that HNRNPM knockdown significantly improves cell survival under low-glucose conditions.

Furthermore, the timing of the decrease in metabolic activity in control cells was determined. No significant reduction in metabolic activity under low-glucose conditions was observed after 36 h, but metabolic activity was found to decrease after 42 h (Figure 4a). In contrast, there was no difference in metabolic activity under high-glucose conditions (Figure 4b).

### 3.3. HNRNPM Knockdown Reduces Glucose Consumption of MIA PaCa-2 Cells by Altering Glucose Metabolism

The glucose concentration in the medium was determined to investigate whether the cell survival benefits conferred by HNRNPM knockdown were due to an alteration of glucose metabolism. We found that the glucose concentration in the medium decreased in a time-dependent manner in all cells but was significantly higher in HNRNPM knockdown cells than in control cells, with a significant difference being observed after 24 h (Figure 5a). Additionally, the lactate concentration in the medium was also found to increase in all cells in a time-dependent manner but was found to be significantly lower in HNRNPM knockdown cells than that in control cells (Figure 5b). Analysis of glucose uptake revealed that HNRNPM knockdown significantly suppressed glucose uptake when compared to that in control cells (Figure 5c). These results indicate that reduced HNRNPM expression alters glucose metabolism and prolongs cell survival by reducing glucose consumption under low-glucose conditions.

## 4. Discussion

HNRNPM appears as a cluster of four proteins, namely M1–M4, with a molecular weight of 64–68 kDa in two-dimensional gel electrophoresis. Unlike other heterogeneous nuclear ribonucleoproteins, it has an unusual hexapeptide-repeat region which is rich in methionine and arginine residues. HNRNPM binds directly to nascent RNA polymerase II transcripts via its lysine and arginine residues and is important for RNA-protein interactions, thereby playing an important role in the processing of heterogeneous nuclear RNAs for the formation of mature mRNAs, as well as in regulating gene expression [7,18,19,20,21,22]. Indeed, HNRNPM was found to be expressed in various normal tissues, of which the pancreas demonstrated the highest HNRNPM levels (Figure 1a). Therefore, it was suggested that the pancreatic tissue might depend largely on HNRNPM-induced regulation of gene expression. Previous studies have reported that HNRNPM upregulation is correlated with poor outcomes and chemoresistance in sarcoma patients, as well as with poor outcomes in colorectal cancer patients [8,10,11]. Moreover, HNRNPM was also found to promote tumor growth, invasion, and metastasis in breast and colon cancer [12,13,14,15,16]. In contrast, HNRNPM expression has been reported to be significantly lower in prostate cancer than that in normal prostate tissues. Furthermore, HNRNPM downregulation was found to be correlated with more aggressive characteristics of prostate cancer cell lines, while low HNRNPM expression levels were shown to promote invasion and migration in prostate cancer [17]. Therefore, this suggests that alterations in HNRNPM expression differ depending on the cancer site and that its effects also vary depending on the cancer type.

In this study, we found that the expression of HNRNPM was downregulated in PDA tissues (Figure 1b). However, the small number of samples analyzed is a limitation of this study. These results differ from those obtained for other types of cancers, with the exception of prostate cancer, and may be associated with the hypovascular environment, which is a characteristic of pancreatic cancer. The hypovascular environment promotes ischemia where blood flow and nutrition are poor, and pancreatic tumors have been shown to be resistant to this type of environment [3,5,6].

Furthermore, Zarei et al. have reported that aggressive tumor characteristics may arise due to selective pressure in a harsh hypovascular environment that is deficient in nutrients and promotes chemotherapy resistance of PDAs under low-glucose conditions [23]. Thus, prolongation of cell survival in this environment has been suggested to play an important role in the progression of PDAs. Indeed, knockdown of HNRNPM protein expression was shown to significantly prolong cell survival under low-glucose conditions (Figure 3a,b). Interestingly, the Cell Counting Kit-8 assay, which measures metabolic activity, and the CellTiter-Glo 2.0 Cell Viability Assay, which measures the amount of ATP, showed different results at 48 h (Figure 3a,b). However, micrographs showed a marked increase in the number of cells after 60 h when compared to that obtained after 48 h (Figure 3c), which was consistent with the amount of ATP determined. Therefore, this indicates that metabolic activity is reduced by nutrient depletion before a decrease in the number of cells occurs. Furthermore, no significant reduction in metabolic activity was observed in control cells under low-glucose conditions after 36 h but was observed after 42 h (Figure 4a). In contrast, there was no difference in metabolic activity observed under high-glucose conditions (Figure 4b). Thus, these results indicate that the decrease in metabolic activity depends on the concentration of glucose.

Furthermore, knockdown of HNRNPM protein expression was shown to significantly increase glucose levels in the medium (Figure 5a). This indicates that knockdown of HNRNPM protein expression suppressed glucose consumption and caused a shift to a more efficient energy production condition. Mitochondrial oxidative phosphorylation (OXPHOS) is reportedly the major pathway necessary for the optimal growth of cancers, including PDAs, under low-glucose conditions [24], and its growth is restricted by glucose levels rather than oxygen levels in the microenvironment [3]. Oxygen levels in the tumor microenvironment are less than 2% but mitochondria function normally even at 0.5% oxygen [25,26,27]. Moreover, we also found that knockdown of HNRNPM protein expression significantly reduced lactic acid levels in the medium and significantly suppressed glucose uptake (Figure 5b,c), suggesting an increase in the ratio of OXPHOS. Therefore, these results suggest that the decrease in HNRNPM expression in PDAs may be involved in tumor adaptation to a hypovascular environment via alterations in glucose metabolism.

In conclusion, in this study, we revealed that HNRNPM was highly expressed in pancreatic tissues but expression decreased in PDA tissues. Furthermore, we found that knockdown of HNRNPM protein expression under low-glucose conditions altered glucose metabolism and prolonged cell survival by suppressing glucose consumption. In the future, comprehensive analysis of gene changes, occurring due to the decreased expression of HNRNPM, and investigation of the detailed mechanism underlying its effects on glucose metabolism are warranted. In addition, it also is necessary to examine tumor growth and cancer invasion and metastasis by the decreased expression of HNRNPM using the 3D culture and xenograft model.

## Figures and Tables

**Figure 1 biology-10-00057-f001:**
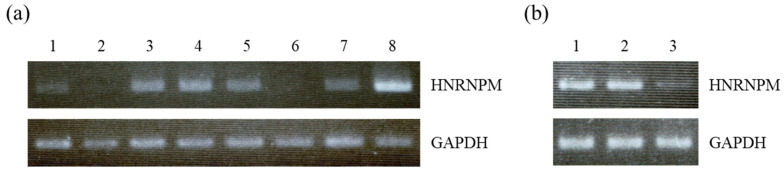
Expression of HNRNPM mRNA in various normal tissues, as well as normal and diseased pancreatic tissues. HNRNPM mRNA levels were determined by PCR. GAPDH mRNA expression served as a quantitative control. (**a**) Lane 1: heart, 2: brain, 3: placenta, 4: lung, 5: liver, 6: skeletal muscle, 7: kidney, 8: pancreas. (**b**) Lane 1: healthy subjects, 2: patients with diabetes, 3: pancreatic ductal adenocarcinoma (PDA) patients.

**Figure 2 biology-10-00057-f002:**
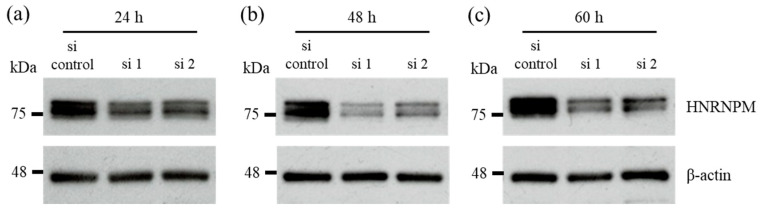
Knockdown of HNRNPM protein expression. Cells were pre-treated with an siRNA against HNRNPM or a non-targeting siRNA and incubated for 24 h (**a**), 48 h (**b**), and 60 h (**c**) under low-glucose conditions. The effect of the knockdown of HNRNPM protein expression was detected using western blotting. si control: non-targeting siRNA-transfected cells, si 1: HNRNPM siRNA (SASI_Hs01_00049126)-transfected cells, and si 2: HNRNPM siRNA (SASI_Hs01_00049128)-transfected cells.

**Figure 3 biology-10-00057-f003:**
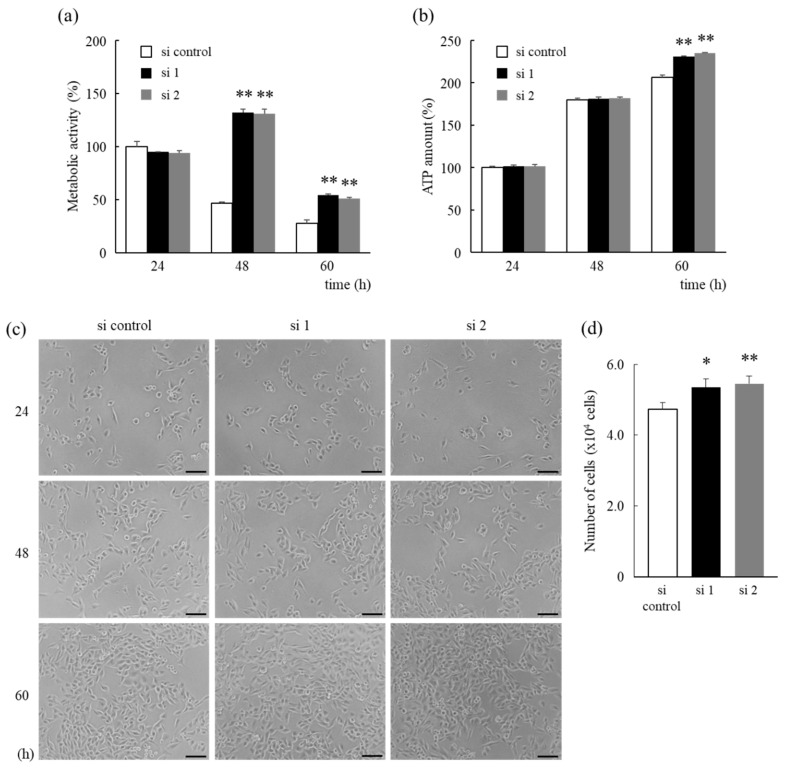
Extended cell survival under low-glucose conditions due to HNRNPM knockdown. (**a**–**c**) Cells were pre-treated with an siRNA against HNRNPM or a non-targeting siRNA and incubated for 24 to 60 h under low-glucose conditions. Cell proliferation was determined using the Cell Counting Kit-8 assay (**a**) metabolic activity) and the CellTiter-Glo 2.0 Cell Viability Assay (**b**) ATP amount). (**c**) Cells were photographed using a differential-phase microscope with 100 X magnification (scale bar, 100 μm). (**d**) Cells were pre-treated with an siRNA against HNRNPM or a non-targeting siRNA and incubated for 60 h. The number of cells was determined via cell counting. Data are shown as mean ± SD (n = 3); *, *p* < 0.05; **, *p* < 0.01 compared to each si control. si control: non-targeting siRNA-transfected cells, si 1: HNRNPM siRNA (SASI_Hs01_00049126)-transfected cells, and si 2: HNRNPM siRNA (SASI_Hs01_00049128)-transfected cells.

**Figure 4 biology-10-00057-f004:**
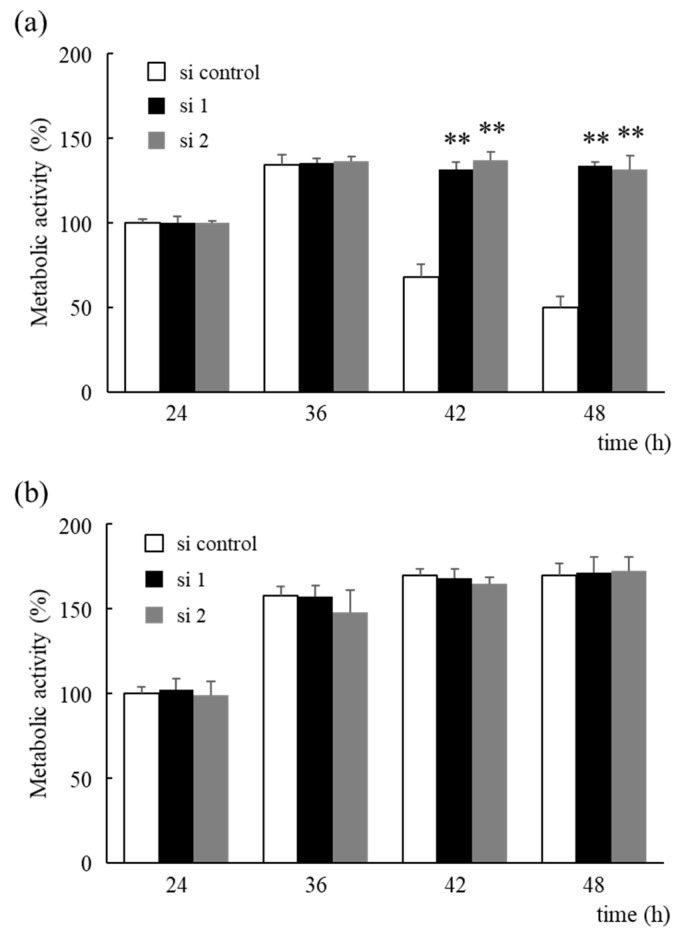
Alterations in metabolic activity due to HNRNPM knockdown under low-glucose conditions. (**a**,**b**) Cells were pre-treated with an siRNA against HNRNPM or a non-targeting siRNA and incubated for 24 to 48 h. Metabolic activity was determined using the Cell Counting Kit-8 assay under low-glucose (**a**) or high-glucose conditions (**b**). Data are shown as mean ± SD (n = 3); **, *p* < 0.01 compared to each si control. si control: non-targeting siRNA-transfected cells, si 1: HNRNPM siRNA (SASI_Hs01_00049126)-transfected cells, and si 2: HNRNPM siRNA (SASI_Hs01_00049128)-transfected cells.

**Figure 5 biology-10-00057-f005:**
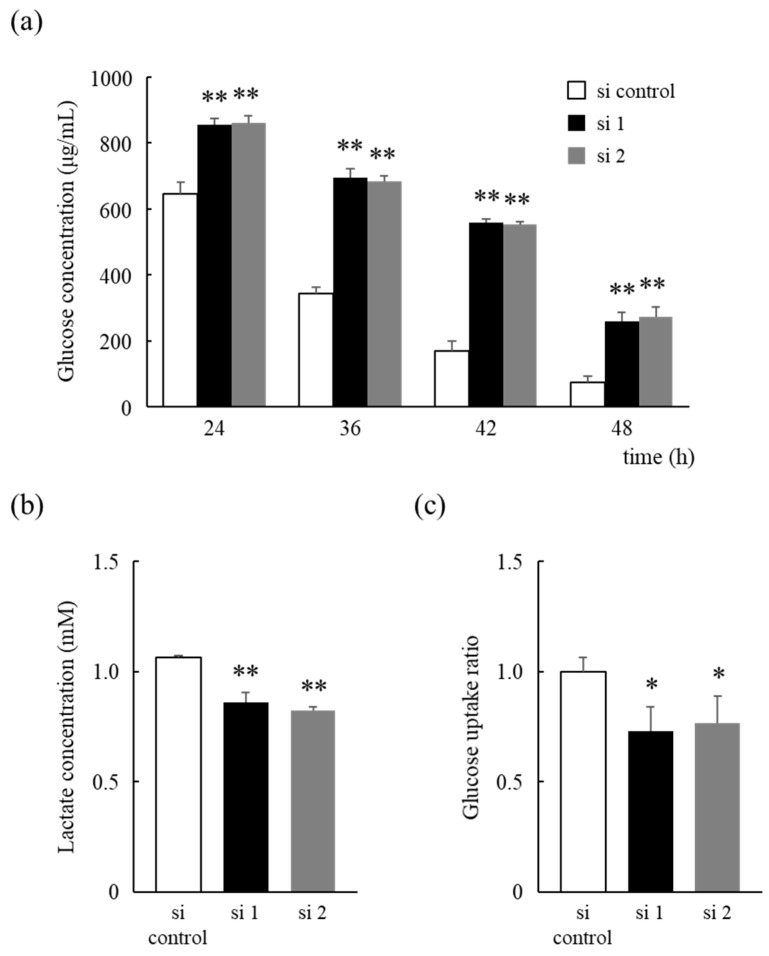
Suppression of glucose consumption and lactate production in HNRNPM knockdown cells. (**a**) Cells were pre-treated with an siRNA against HNRNPM or a non-targeting siRNA and incubated for 24 to 48 h under low-glucose conditions. The glucose concentration in the medium was determined using the Glucose kit Glucose C II Test Wako. (**b**,**c**) Cells were pre-treated with an siRNA against HNRNPM or a non-targeting siRNA, and incubated for 48 h under low-glucose conditions. The lactate concentration in the medium was determined using the Glycolysis Cell-Based Assay Kit (**b**). Glucose uptake was determined using the Glucose Uptake-Glo Assay and the CellTiter-Glo 2.0 Cell Viability Assay (**c**). Data are shown as mean ± SD (n = 3); *, *p* < 0.05; **, *p* < 0.01 compared to each si control. si control: non-targeting siRNA-transfected cells, si 1: HNRNPM siRNA (SASI_Hs01_00049126)-transfected cells, and si 2: HNRNPM siRNA (SASI_Hs01_00049128)-transfected cells.

## Data Availability

The data presented in this study are available on request from the corresponding author.

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
