# Peer review of "Alterations in Glucose Metabolism Due to Decreased Expression of Heterogeneous Nuclear Ribonucleoprotein M in Pancreatic Ductal Adenocarcinoma"

_biology, 2021, doi:10.3390/biology10010057_

Round 1

Reviewer 1 Report

In the manuscript entitled “Alterations in glucose metabolism due to decreased expression of heterogeneous nuclear ribonucleoprotein M in pancreatic ductal adenocarcinoma”, the authors present  the role of HNRNPM in glucose metabolism and survival of pancreatic ductal adenocarcinoma.

The authors demonstrate the impact that the knockdown of HNRNPM protein has on PDAC cell growth and metabolism in vitro. The results presented by the authors support their hypothesis, however, the quality of the work would greatly  improve if the following issues would be considered:

  • In the introduction, the same information, even the same sentences are repeated. The authors should expand on the information a little and add more information, compared to the abstract. Also the sentences in the lines 50-52 and 58-60 are a bit unclear and the parts of the sentence are not related.
  • In the methods section, could the authors add more information? What concentration of siRNA is used for the experiment, what plate formats are used etc.
  • The authors state that the expression of HNRNPM is lower in PDAC samples compared to normal tissues based on one sample. This is not representative. Could the authors analyse more tissue samples or present some database search analysis that would support that ?
  • The authors present most of the experiment in the low glucose conditions. However, PDAC is characterized also by low oxygen supply. Did the authors consider performing the experiments in the low glucose/hypoxic conditions ?
  • The authors performed all of the experiments in one cell line, MiaPaCa. Whas there any specific reason for choosing this cell line ? Could the authors verify if the same effects would be achieved in other cell lines ?
  • The authors say that the metabolic activity was nit changed until 36 h after transfection, but the changes were noted after 42 h. Could the authors expand on that ? What may be the reason ?
  • Based on the glucose and lactate consumptions, the authors suggest that the HNRNPM influences the OXPHOS. Could the authors actually verify that, e.g with the use of the Seahorse XF ?
  • The authors show the increase in cell number after the knockdown of HNRNPM, however, the increase is minimal in the used conditions, Did the authors consider 3D cultures? It would be more informative.
  • In addition, stable knockdown of HNRNPM and the verification of its impact on tumour growth in the xenograft model would be a valuable addition to the study.

Reviewer 2 Report

The results presented in this article are noteworthy and represent an advancement in our understanding of pancreatic cancer. 

This work is brief and communicates the methods and results effectively.  There is a tendency to discuss the results and draw conclusions in the results section of the paper, rather than in the discussion.  The figures are easy to read, but any figure in which time is expressed has the units shown as (h) off to the side of the figure.  Perhaps a label such as "time / h" or "time (h)" centered on the appropriate axis would make it easier to understand the units?

The methods seem clearly defined and other should be able to reproduce the investigation.

Overall, this work is scientifically and methodologically sound, but some revision to the presentation is in order.

Round 2

Reviewer 1 Report

I would like to thank the authors of the manuscript entitled "Alterations in glucose metabolism due to decreased expression of heterogeneous nuclear ribonucleoprotein M in pancreatic ductal adenocarcinoma" for implementing some of the comments in the presented manucsript.

As an overall remark I would like to recommend to the authors to always use more than one cell line in all the experiments done to increase the credibility of the data and to minimize the possibility that the results seen in one cell line are due to some mutations specific only to this cell line and would not be reproduced in others.